# Faster MDNet for Visual Object Tracking

**Qianqian Yu** [ID]**, Keqi Fan, Yiyang Wang and Yuhui Zheng** *[ID]

Engineering Research Center of Digital Forensics, Ministry of Education,
Nanjing University of Information Science and Technology, Nanjing 210044, China;
20201220051@nuist.edu.cn (Q.Y.); 20201220013@nuist.edu.cn (K.F.); 201883290297@nuist.edu.cn (Y.W.)
* Correspondence: zheng_yuhui@nuist.edu.cn

**Abstract:** With the rapid development of deep learning techniques, new breakthroughs have been made in deep learning-based object tracking methods. Although many approaches have achieved state-of-the-art results, existing methods still cannot fully satisfy practical needs. A robust tracker should perform well in three aspects: tracking accuracy, speed, and resource consumption. Considering this notion, we propose a novel model, Faster MDNet, to strike a better balance among these factors. To improve the tracking accuracy, a channel attention module is introduced to our method. We also design domain adaptation components to obtain more generic features. Simultaneously, we implement an adaptive, spatial pyramid pooling layer for reducing model complexity and accelerating the tracking speed. The experiments illustrate the promising performance of our tracker on OTB100, VOT2018, TrackingNet, UAV123, and NfS.

**Keywords:** visual object tracking; deep learning; domain adaptation

## 1. Introduction

As one of the fundamental tasks in computer vision, object tracking methods use the contextual information of video sequences to model object appearance and motion information for predicting object positions and motion states. Civil and military systems based on video object tracking have landed and are widely employed in important fields, such as intelligent transportation, autonomous driving, unmanned aerial vehicles, human–machine interaction, and radar tracking.

Since AlexNet [1] made a large splash in image classification tasks, deep learning has gained widespread attention. With a powerful feature extraction capability and end-to-end training models, deep learning techniques have made great progress in computer vision, machine learning, natural language processing, and other fields. In the past few years, deep learning-based object tracking algorithms have made significant breakthroughs. The initial deep tracking methods focused on correlation filters, which replace manual features in traditional correlation filters with deep features or combine end-to-end training of deep networks with correlation filters. HCF [2] is proposed to separately train correlation filters for deep features of different layers and perform feature fusion to better utilize the deep model. Subsequently, HDT [3] adaptively changes the weights of filters at different scales, and MCCT [4] combines various features of filters and switches them adaptively.

Based on deep correlation filter tracking methods, C-COT [5] focused on the problem of response map fusion caused by differences in the resolution of deep features at different scales. Inspired by C-COT, ECO [6] performs adaptive correlation filter selection, clustering of target samples, and sparse target updating to obtain further optimization in terms of efficiency and storage. Despite the excellent performance achieved by the above-mentioned deep correlation tracking methods on several datasets, the time-consuming extraction of features keeps them from achieving real-time speed even when running on an advanced GPU.

In 2016, Bertinett et al. proposed the Siamese network SiamFC [7], which extracts features from the target template and search region by using a convolutional network and then generates the response map. The peak point on the response map is the target position. Considering that the regression of the target scale by SiamFC in a traditional scaling form still cannot accurately obtain scale information of the target, Li et al. presented SiamRPN [8]. This method introduces the RPN [9] structure from object detection to SiamFC. Features are extracted by using the shared parameters. The location of the target is obtained through a classification branch, and an accurate estimate of the target scale is obtained by the regression branch. Li et al. further proposed DaSiamRPN [10], which improves the anti-interference ability of the tracker by effectively using negative samples. Nevertheless, AlexNet [1] is employed as the feature extraction network in the abovementioned methods instead of the more robust models that are currently utilized. SiamRPN++ [11] modifies the ResNet [12] network to obtain promising results on several datasets. Ocean [13] used an anchor-free regression network based on SiamFC [7] to regress the object region over a large space. Despite the great success of the abovementioned approaches, the lack of an online update process for these methods causes the problem that trackers cannot automatically adjust parameters to suit changes in objects.

ATOM [14] uses a conjugate gradient strategy combined with a deep learning framework for jointly fast optimization. The authors further employed the idea in DiMP [15] for end-to-end learning to achieve leading performance by learning all parameters through neural networks. Based on DiMP, PrDiMP [16] uses a probabilistic regression-based approach to model the tracker and scale prediction modules, which alleviates the interference of imprecise or ambiguous noise labels on the network and effectively improves tracking accuracy. Due to the use of complicated deep learning network structures, these methods typically require a large storage space.

Conventional trackers generally treat the tracking task as frame-by-frame object detection while disregarding the abundant temporal relations. Wang et al. [17] explored temporal information in videos by utilizing the Transformer structure to bridge independent video frames. TransT [18] draws on the Transformer structure for improving feature fusion operations in traditional two-way networks. TransT also employs the attention mechanism from Transformer to fuse the template information into the search region for better object localization and scale regression. SwinTrack [19] allows full interaction between the object and the search region, as well as a comprehensive study of different strategies for feature fusion, location encoding, and training loss to further improve model performance. These approaches employ deep CNN models, such as ResNet 50 [12]. Sophisticated models consume tens or even hundreds of megabytes of memory, limiting their practical application.

MDNet [20] is the first lightweight tracker to introduce classification-based networks into the field of tracking. To address the problem that a target in one video may become part of the background in other videos, a training mode under multiple target domains is introduced. VITAL [21] implements a generative adversarial network in MDNet to interfere with classifiers by generating masks with occlusion properties, making the features learned by classifiers more robust. RT-MDNet [22] introduces ROI Align [23] to MDNet, which improves the tracking speed. However, the large number of features to be extracted limits the tracking speed of these trackers.

Based on these findings, we propose a novel network structure, Faster MDNet, to seek a balance among tracking accuracy, speed, and computational resource consumption. First, we introduce a channel attention module after convolutional layers to implement a strategy for capturing cross-channel interactions using fast one-dimensional convolution. Second, redundant information is effectively removed, and more generic features are transferred to our model by domain adaptation components. Last, we design an adaptive spatial pyramid pooling layer. Since feature maps of different sizes are input to this layer and will become the same size, this layer can be utilized as a substitute for fully connected layers.

Our contributions are summarized as follows:

- We introduce a channel attention module after convolutional layers. Model robustness is improved by fast one-dimensional convolution without dimensionality reduction while adding only a few parameters. We use the ILSVRC dataset from the image classification domain as training data and design domain adaptation components for transferring feature information. The adaptive spatial pyramid pooling layer with a multi-scale feature fusion strategy is implemented to reduce model complexity and to improve model robustness.

- Our model has a simple structure that enables lightweight tracking. Due to a significant reduction in model complexity, the proposed tracker has a high real-time speed. The tracker uses both high-level semantic information and low-level semantic information for a higher tracking accuracy.

The experimental results obtained with several popular datasets demonstrate that our approach has a performance similar to that of current state-of-the-art trackers while consuming fewer computational resources.

We note that a shorter conference version of this paper appeared in Progress in Informatics and Computing (2021). The proposed method in our initial conference paper did not achieve a tracking accuracy comparable to that of state-of-the-art trackers. This manuscript addresses this issue and provides additional analysis on the attention mechanism. Experiments were conducted with more datasets.

The remainder of this paper is organized as follows: In Section 2, domestic and international studies related to our approach are illustrated. Section 3 describes our network in detail and gives an overview of the tracking process. The fourth section shows the experimental results obtained with different datasets, such as OTB100 [24], VOT2018 [25], TrackingNet [26], UAV123 [27], and NfS [28]. We draw conclusions in Section 5.

## 2. Related Work

### 2.1. MDNet-Based Trackers

Inspired by R-CNN [29] from object detection, MDNet [20] treats object tracking tasks as a task of distinguishing targets and backgrounds. MDNet contains shared layers for generic feature extraction and domain-specific layers with classifiers. Each domain-specific layer corresponds to a video sequence. Such domain-specific layers are updated online during the tracking process. Although MDNet achieves excellent results with generic datasets, it achieves a tracking speed of 1 frame per second (FPS), failing to meet the requirements of real-time tracking. VITAL [21] employs the generated adversarial network into MDNet. Through adversarial learning, this generative network allows classifiers to learn more robust features, thereby suppressing masking and preventing the model from overfitting. VITAL effectively improves tracking accuracy but does not even run at 1 FPS on GPU. RT-MDNet [22] draws on the idea of Fast R-CNN [30] to extract shared features of the search region and to improve object localization using the ROI Align. The tracking speed of this approach is greatly improved but limits the tracking accuracy. Considering that the idea of these methods has a certain connection with image classification tasks, we make enhancements on MDNet to improve both tracking accuracy and speed.

### 2.2. Attention Mechanism in Computer Vision Tasks

To make reasonable use of limited visual information processing resources, methods for selectively focusing on the most important regions of an image while disregarding other visible information are referred to as attention mechanisms. Attention mechanisms have been widely employed in computer vision tasks, such as image classification, object detection, face identification, few-shot learning, image generation, and object tracking since they allow for more efficient visual information processing.

One of the representative works is SENet [31], which includes a squeeze module and an excitation module. SENet adaptively adjusts the feature responses between two channels by means of feature rescaling to emphasize important feature maps. The SE module can be easily implemented in current network structures to improve performance.

Nevertheless, the squeeze module is too simplistic to capture sophisticated global information. Simultaneously, fully-connected layers in the excitation module increase the model complexity. Subsequent studies revolve around these problems. Gao et al. [32] introduced a global second-order pooling (GSoP) module to improve the squeeze module from SENet by efficiently capturing global contextual information. However, the GSoP module can only be added after residual blocks due to the additional computational resource consumption. CBAM [33] combined channel attention and spatial attention, which improved computational efficiency by decoupling attention maps between channel dimensions and spatial dimensions. Spatial global information is utilized by global pooling. ECA [34] proposed a local cross-channel interaction strategy without dimensionality reduction based on SENet. Not only is it beneficial to improve model performance, but it also has a smaller number of parameters compared to other attention modules. For better performance and lower computational resource consumption, we try to implement the ECA module into our model.

## 3. Method

### 3.1. Network Structure

As shown in Figure 1, our model consists of five parts. Convolutional layers of the feature extraction part are derived from VGG-M [35] and are utilized to extract generic features. The channel attention module improves tracking accuracy by capturing local cross-channel interactions. More robust features from the image classification domain are transferred by domain adaptation components. The adaptive spatial pyramid pooling layer is a substitute for the first two fully connected layers of MDNet. The target classification part has K branches, each of which has a binary classification layer with a softmax cross-entropy loss that distinguishes the target from the background. The remainder of this section describes the various parts of our method and shows the complete tracking process.

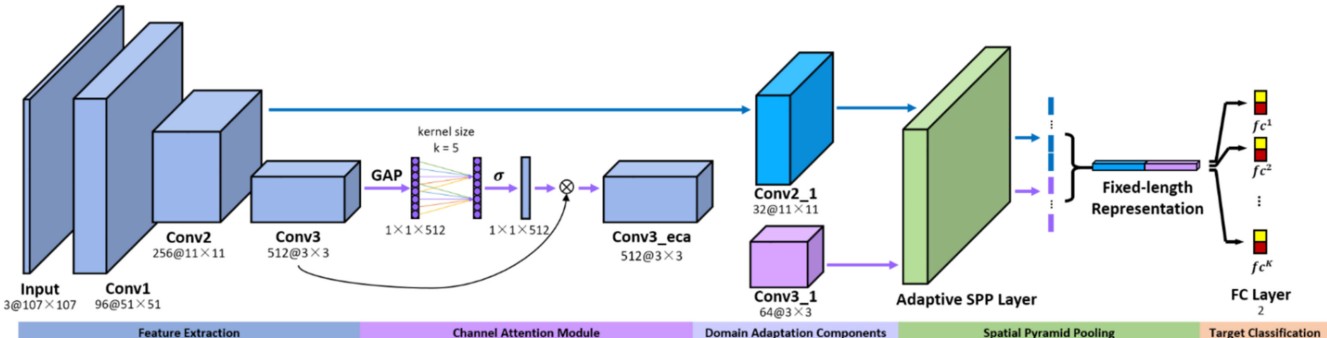

**Figure 1.** Network structure of the proposed method. It consists of feature extraction, a channel attention module, domain adaptation components, spatial pyramid pooling, and target classification.

### 3.2. Channel Attention Module

#### 3.2.1. Local Cross-Channel Interactions

The input feature map is $\mathcal{F} \in \mathbb{R}^{W \times H \times C}$, where $W$, $H$, and $C$ represent the width, height, and channel, respectively. The aggregated feature $\mathbf{y} \in \mathbb{R}^C$ can be computed as:

$$\mathbf{y} = g(\mathcal{F}) \tag{1}$$

where $g(\mathcal{F}) = \frac{1}{WH} \sum_{i=1,j=1}^{W,H} \mathcal{F}_{i,j}$ is channel-wise global average pooling.

In the case of $\mathbf{y} \in \mathbb{R}^C$ without dimensionality reduction, channel attention can be calculated as follows:

$$\omega = \sigma(\mathbf{W}\mathbf{y}) \tag{2}$$

where $\sigma$ is a sigmoid function and $\mathbf{W}$ is a band matrix that has $C \times k$ parameters, i.e.,

$$
\begin{bmatrix}
w^{1,1} & \cdots & w^{1,k} & 0 & 0 & \cdots & \cdots & 0 \\
0 & w^{2,2} & \cdots & w^{2,k+1} & 0 & \cdots & \cdots & 0 \\
\vdots & \vdots & \vdots & \vdots & \ddots & \vdots & \vdots & \vdots \\
0 & \cdots & 0 & 0 & \cdots & w^{C,C-k+1} & \cdots & w^{C,C}
\end{bmatrix}
\tag{3}
$$

For the previously mentioned matrix, the weight of $y_i$ can be computed by the interactions of $y_i$ and $k$ channels adjacent to $y_i$ as:

$$
\omega_i = \sigma\left( \sum_{j=1}^{k} w^j y_i^j \right), y_i^j \in \Omega_i^k
\tag{4}
$$

We implement this strategy by a fast 1D convolution whose kernel size is $k$:

$$
\omega = \sigma(\text{Conv1D}_k(\mathbf{y}))
\tag{5}
$$

The output feature map $Y$ of our channel attention module can be computed as:

$$
Y = \omega \mathcal{F}
\tag{6}
$$

This channel attention module of our work, which only involves $k$ parameters, maintains tracking efficiency.

### 3.2.2. Adaptive Selection of Channel Interactions

Since the channel attention module is designed for capturing local cross-channel interactions, it is necessary to determine the coverage of interactions; that is, the kernel size of fast 1D convolution. The kernel size can be determined by manual tuning but requires a large number of computational resources. In this paper, we use cross-validation to adaptively determine the kernel size. The coverage of the cross-channel interaction is proportional to the channel dimension, i.e., there may be a mapping between $k$ and $C$. We introduce an exponential function to represent the nonlinear mapping relationship as follows:

$$
C = 2^{(\lambda * k - b)}
\tag{7}
$$

The kernel size $k$ can be determined by

$$
k = \varphi(C) = \left| \frac{\log_2 C}{\lambda} + \frac{b}{\lambda} \right|_{odd}
\tag{8}
$$

where $|x|_{odd}$ denotes the nearest odd number of $x$. We set $\lambda$ to 2 and $b$ to 1. It can be seen that low-dimensional channels have a smaller interaction range, while high-dimensional channels have a larger interaction range using mapping $\varphi$.

### 3.3. Domain Adaptive Components

Domain adaptation algorithms employ samples from the source domain that are labeled to optimize the model of the target domain. The learning function is:

$$
f^* = \underset{f \in \mathcal{H}}{argmin} \frac{1}{B} \sum_{i=1}^{B} l(f(x_i), y_i) + \lambda R(\mathcal{B}_s, \mathcal{B}_t)
\tag{9}
$$

where $f(\cdot)$ is the object function; that is, the probability of $x_i$ being a target. B denotes the quantity of samples in a mini-batch. $x_i$ represents the $i$-th sample in training data, and $y_i$ is the label of $x_i$. $y_i$ equals 1 when $x_i$ is the target. $l(\cdot)$ is a loss function, and $\lambda$ is a regularization coefficient. $R(\cdot)$ is a regularization function that measures the complexity

of the model. $\mathcal{B}_s$ and $\mathcal{B}_t$ indicate the samples in the source domain and target domain, respectively.

The convolutional layers in our model draw from VGG-M, which is a model that is widely employed in image classification methods. General features can be extracted by pretrained convolutional layers. In this work, we use ILSVRC [36], a dataset from the image classification domain, to pretrain our model. Image classification tasks concentrate on identifying the category of an object, whereas tasks in the object tracking domain focus on distinguishing the object and background. As a result, we design domain adaptation components for transferring features from the classification domain to the tracking domain to achieve a more robust tracking model. These components treat individual objects as learning targets rather than categories of images. Additionally, our model can remove redundant information to reduce computational complexity by adding domain adaptation components.

### 3.4. Adaptive Spatial Pyramid Pooling Layer with Multi-Scale Feature Fusion Strategy

A CNN network generally consists of convolutional layers and fully connected layers. The former has no requirements for a fixed size of input vectors, yet the latter needs vectors fixed in size. Hence, only images of the same size can be applied as inputs for models. A spatial pyramid pooling layer, which is a pooling structure between the last layer of convolutional layers and the fully connected layer, was proposed to address this challenge. By pooling operations in the spatial pyramid pooling layer, feature maps of different sizes can transform into the same size.

We design an adaptive spatial pyramid pooling layer with a multi-scale feature fusion strategy for aggregating feature maps of different sizes. Moreover, the adaptive spatial pyramid pooling layer can be a substitute for part of the fully connected layers in MDNet [20]. Not only can the spatial pyramid pooling layer reserve semantic information extracted from convolutional layers, but our model can also be more robust by parameter reduction.

Because adding the adaptive spatial pyramid pooling layer after Conv3_eca will result in missing features, we employ feature maps from Conv2 and Conv3_eca as inputs to the fully connected layer. Utilizing low-level features with high resolution and high-level features with more semantic information can improve the tracking accuracy of our model.

### 3.5. Tracking Process

First, the proposed model generates a domain-specific layer for a new test video sequence by accomplishing multi-domain learning. Second, the bounding box regression model is trained with the original frame. We also fine-tuned the domain-specific layer and adaptive spatial pyramid pooling layer and gained the optimal object state. The implementation of our tracking method is presented in Algorithm 1. Details of the tracking process are discussed in this section.

#### 3.5.1. Multi-Domain Learning

Videos in the training data are derived from different domains, and every video contains domain-specific feature information. Nevertheless, general information, i.e., the robustness of scale variations, illumination variations, and motion blur, exists in different video sequences. Hence, we implement our model with a multi-domain learning strategy to separate domain-specific information from general information. We train our model by stochastic gradient descent (SGD) and individually process the video of each domain in each iteration. In the $t$-th iteration, our model is updated by a minibatch that consists of training samples from the $(t \bmod K)$-th video sequence. Only the $(t \bmod K)$-th fully connected layer is enabled. Iterations are repeated unless the model converges or the predefined number of iterations is achieved. We compare samples generated in each frame of video sequences with the ground truth. Fifty positive samples with an intersection over union (IoU) value higher than 0.7 and two hundred negative samples with an IoU value lower than 0.5 are selected.

---

**Algorithm 1.** Tracking Algorithm

---

**Input:** Pretrained filters $\{m_1, m_2, m_{2\_1}, m_3, m_{3\_eca}, m_{3\_1}, m_{spp}\}$
Primary target position $c_1$
**Output:** Optimal target positions $c_t^*$
1: Randomly initialize the fully-connected layer $m_{fc}$.
2: Use $\Phi_{3\_eca}$ to train a bounding box regression model.
3: Generate positive samples $S_1^+$ and negative samples $S_1^-$.
4: Update $\{m_{spp}, m_{fc}\}$ by $S_1^+$ and $S_1^-$.
5: $\mathcal{U}_s \leftarrow \{1\}$ and $\mathcal{U}_l \leftarrow \{1\}$.
6: Optimal target positions $c_t^*$
7: Randomly initialize the fully-connected layer $m_{fc}$.
8: **repeat**
9:         Generate target candidate samples $c_t^i$.
10:        Obtain the optimal target state $c_t^*$ by (10).
11:      **if** $f^+(c_t^*) > 0.5$ **then**
12:              Plot training samples $S_t^+$ and $S_t^-$.
13:              $\mathcal{U}_s \leftarrow \mathcal{U}_s \cup \{t\}$ and $\mathcal{U}_l \leftarrow \mathcal{U}_l \cup \{t\}$
14:              **if** $|\mathcal{U}_s| > \mathcal{U}_s$ **then** $\mathcal{U}_s \leftarrow \mathcal{U}_s \backslash \{min_{x \in \mathcal{U}_s} x\}$.
15:              **if** $|\mathcal{U}_l| > \mathcal{U}_l$ **then** $\mathcal{U}_l \leftarrow \mathcal{U}_l \backslash \{min_{x \in \mathcal{U}_l} x\}$.
16:              Fit $c_t^*$ by bounding box regression model.
17:      **if** $f^+(c_t^*) < 0.5$ **then**
18:              Update $\{m_{spp}, m_{fc}\}$ using $S_{x \in \mathcal{U}_s}^+$ and $S_{x \in \mathcal{U}_s}^-$.
19:      **else if** t mod 10 = 0 **then**
20:              Update $\{m_{spp}, m_{fc}\}$ using $S_{x \in \mathcal{U}_l}^+$ and $S_{x \in \mathcal{U}_s}^-$.
21: **until end of sequence**

---

### 3.5.2. Optimal Target Position Estimation

To estimate the target state from each frame in the video sequence, we sample the previous frame with a multidimensional Gaussian distribution in width, height, and channel to obtain $N$ (=256) candidates and then calculate their positive and negative scores. The optimal object state is given by:

$$c^* = \underset{c^i}{\operatorname{argmax}} f^+\left(c^i\right) \tag{10}$$

where $f^+(\cdot)$ is the probability that the sample is the object.

### 3.5.3. Bounding Box Regression

We train a bounding box regression model by the first frame of a video to solve the target location drift problem. Input feature vectors, and the predicted values are obtained by translation and scaling operations. The objective function can be defined as:

$$d_*(c) = w_*^T \phi_{3_{eca}}(c) \tag{11}$$

where $w_*^T$ is a parameter to be learned (* represents X, Y, W, and H. X denotes the x-coordinate; Y indicates the y-coordinate; W represents the weight, and H represents the height. Each dimensional transformation corresponds to an objective function), $\phi_{3\_eca}(c)$ is the feature vector obtained from Conv3_eca, and $d_*(c)$ is the predicted value.

### 3.5.4. Network Update

We update our model by means of long-term updates and short-term updates. The former exists in regular iterations and includes $\mathcal{U}_l$ samples, which are executed every 8 frames. The latter, which contains $\mathcal{U}_s$ samples, is executed when the estimated score is less than 0.5.

3.5.5. Loss Function

We use BCE loss for binary classification to optimize our model. The calculation formula is expressed as follows:

$$\mathcal{L}_n = -\left(y_n \cdot \log f^+(c^n) + (1 - y_n) \cdot \left(1 - \log f^+(c^n)\right)\right) \tag{12}$$

where $y_n \in \{0, 1\}$ is the label of the $n$-th sample. If the sample is the target, $y_n = 1$.

## 4. Experiments

### 4.1. Implementation Details

For pretraining, we train the feature extraction part, channel attention module, domain adaptation components, and adaptive spatial pyramid pooling layer of the proposed network on the ILSVRC [36] dataset. The weights of layer $i$ are denoted by $m_i$, where $m_{1:spp}$ are obtained by pretraining. We train convolutional layers by a learning rate of 0.0001 and train the fully connected layer with a learning rate of 0.001. For online updating, we set the number of samples $\mathcal{U}_l$ to 100 and set $\mathcal{U}_s$ to 20. The momentum is set to 0.9 with the weight decay set to 0.0005. Our tracker is implemented in Python by PyTorch. All experiments run with a NIVIDA RTX 2080 ti GPU and an Intel I9-9900X CPU.

### 4.2. Options of Kernel Size k

In this part, we determine the optimal coverage of interactions by adaptive selection of kernel size. Therefore, we train our model with the ECA [34] module by setting $k$ from 3 to 9. As shown in Figure 2, the best result obtained is 69.4% when $k$ is set to 5, which is 0.5% greater than the original results. Since the proposed network has more intermediate layers affecting the performance of the tracking performance, a smaller kernel size may be preferred for our model. The conclusion can be drawn that since manual tuning of $k$ can be avoided by cross-validation, the adaptive selection of kernel size based on (8) tends to perform better than fixing the kernel size.

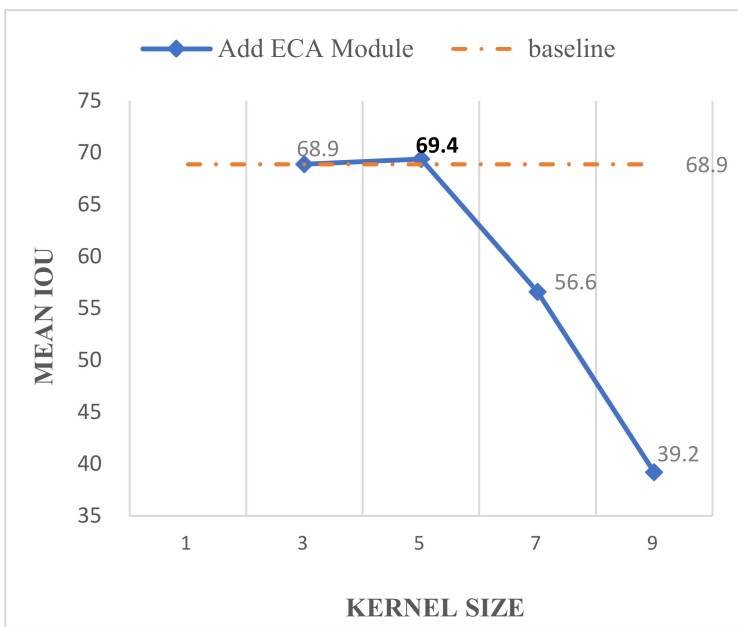

**Figure 2.** Results of our model adding attention modules with different kernel sizes on OTB100. Here, we consider the results without the attention module as the baseline. The tracking accuracy of the model varies with the kernel size.

### 4.3. Comparisons of Different Attention Modules

We apply the ECA [34] module and other state-of-the-art attention modules, i.e., the SE [31], GSoP [32], and CBAM [33] modules, to the proposed network for comparison. We consider our model without adding an attention module as the baseline. Table 1 shows that the proposed model with the ECA module shares almost the same computational complexity as the original network while achieving 0.5% gains in IoU. It can be seen that the network with the GSoP module achieves the highest IoU value while requiring a larger number of FLOPs and parameters. Compared with the SE, GSoP, and CBAM modules, the ECA module obtains better results while consuming fewer computational resources.

**Table 1.** Results of adding different attention modules to our model in terms of parameters, FLOPs, and mean IoU value on OTB100. The model without the attention module is considered the baseline. The best-performing value for each metric is shown in bold.

| Module | Parameters | FLOPs | Mean IoU |
|:---:|:---:|:---:|:---:|
| Baseline | 4,170,370 | 122.14 M | 0.689 |
| +SE Module [31] | 4,203,138 | 122.17 M | 0.674 |
| +GSop Module [32] | 4,480,264 | 252.11 M | **0.695** |
| +CBAM Module [33] | 4,203,236 | 122.17 M | 0.676 |
| +ECA Module [34] | **4,170,375** | **122.14 M** | 0.694 |

### 4.4. Results on OTB100

The OTB100 [24] dataset contains 100 video sequences and takes into account many factors that affect tracker performance. Eleven attributes, such as deformation, occlusion, fast motion, and motion blur, are labeled for video sequences to analyze the ability of trackers to cope with different scenarios. The dataset contains precision plots and success plots to evaluate tracker performance. Precision plots measure trackers by comparing the Euclidean distance between the center of the predicted position and the ground truth. The formula for calculating the precision is given by:

$$Precision(G, P) = \left( \sum_{i=1}^{n} |G_i - P_i|^2 \right)^{1/2} \tag{13}$$

where $i$ is the target dimension, which is typically set to 2. $G_i$ is the center coordinate of the ground truth, and $P_i$ is the center coordinate of the target position predicted by the tracking algorithm.

Success plots are calculated by using the intersection over union (IoU) value to predict the degree of overlap between the target and the ground truth. The calculation formula of success rate is expressed as follows:

$$Success(G, P) = \frac{S_G \cap S_P}{S_G \cup S_P} \tag{14}$$

where $S_G$ is the area of the ground truth and $S_P$ is the area predicted by the tracking method.

#### 4.4.1. Comparisons with State-of-the-Art Trackers

Figure 3 shows that our tracker performs better than SiamRPN++ [11]. Compared with MDNet [20], our method improves the precision by 1.2% and the success rate by 2.1%.

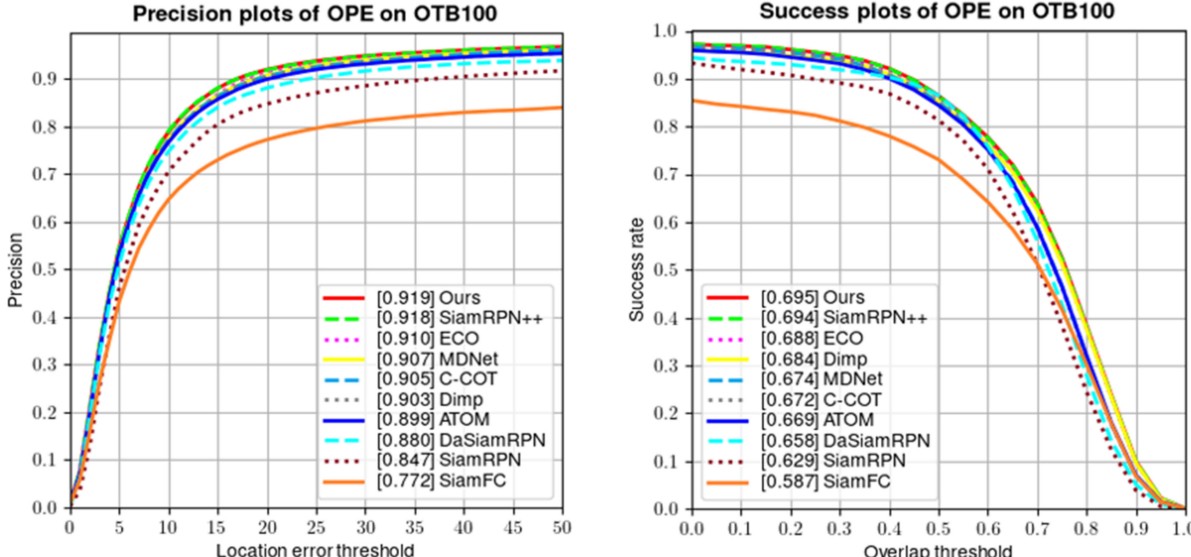

**Figure 3.** Results of current trackers and ours on OTB100 in terms of precision plots and success plots. OTB100 using one-pass evaluation for testing trackers.

### 4.4.2. Comparisons with MDNet-Based Trackers

We also compared our tracker with MDNet [20] and two methods based on MDNet, VITAL [21], and RT-MDNet [22]. As shown in Table 2, our tracker achieves the best results in terms of both precision and AUC (Area-Under-Curve) score. In addition, our tracker achieves a tracking speed of 36 FPS, which is much faster than other methods.

**Table 2.** Results of several MDNet-based methods on OTB100 containing precision, AUC score, and FPS.

| Trackers | Precision | AUC | FPS |
|---|---|---|---|
| MDNet [20] | 0.907 | 0.674 | 1 |
| VITAL [21] | 0.902 | 0.674 | <1 |
| RT-MDNet [22] | 0.868 | 0.657 | 25 |
| Ours-conf | 0.912 | 0.681 | 35 |
| Ours | **0.919** | **0.695** | **36** |

### 4.5. Results on VOT2018

The VOT2018 [25] dataset evaluates the performance of algorithms for not only short-term tracking but also long-term tracking. Each time that a tracker fails to track an object on this dataset, it will restart tracking. The performance of trackers is evaluated based on the accuracy and number of failures. Accuracy refers to the average overlap rate under a test video sequence and is expressed as follows:

$$Accuracy = \frac{1}{N_v} \sum_{t=1}^{N_v} \frac{1}{N_{rep}} \sum_{k=1}^{N_{rep}} \frac{S_G \cap S_P}{S_G \cup S_P} \tag{15}$$

where $t$ is the present number of video frames, $k$ is the $k$-th test of the current measurement, $N_v$ represents the number of valid tracking frames, and $N_{rep}$ is the number of repetitions.

Robustness is the ratio between the number of tracking failures and the number of valid tracking frames. The EAO value reflects the relationship between the length of sequences and the average accuracy.

Table 3 shows comparisons of other state-of-the-art trackers with our tracker. It can be seen that the proposed tracker has the same accuracy value as PrDiMP [16], one of the

most advanced methods. In terms of robustness and Expect Average Overlap (EAO) value, our method achieves 0.432 and can also be competitive with leading trackers.

**Table 3.** Results of our model with other high-performing trackers on VOT2018 with three metrics; accuracy, robustness, and EAO value.

| Trackers | Accuracy | Robustness | EAO |
|---|---|---|---|
| ATOM [14] | 0.59 | 0.20 | 0.401 |
| C-COT [5] | 0.50 | 0.32 | 0.267 |
| DaSiamRPN [10] | 0.59 | 0.28 | 0.383 |
| DiMP [15] | 0.60 | **0.15** | 0.440 |
| ECO [6] | 0.48 | 0.28 | 0.276 |
| MDNet [20] | 0.55 | 0.38 | 0.383 |
| PrDiMP [16] | **0.62** | 0.17 | **0.442** |
| SiamFC [7] | 0.50 | 0.59 | 0.188 |
| SiamRPN [8] | 0.59 | 0.28 | 0.383 |
| SiamRPN++ [11] | 0.60 | 0.23 | 0.414 |
| Ours-conf | 0.60 | 0.21 | 0.409 |
| Ours | **0.62** | 0.16 | 0.432 |

We summarized the EAO value, FLOPs, and parameters of Ocean [13], PrDiMP [16], SiamFC [7], SiamRPN++ [11] and our method on the VOT2018 [25] dataset. To visually compare the performance of trackers, we set the vertical coordinate as the ratio of the EAO value to the number of parameters. The higher value of the vertical coordinate proves the better performance of the method. As shown in Figure 4, only the proposed tracker and SiamRPN++ [11] have a model FLOPs number lower than 10,000 M, and our method has the best cost-effectiveness.

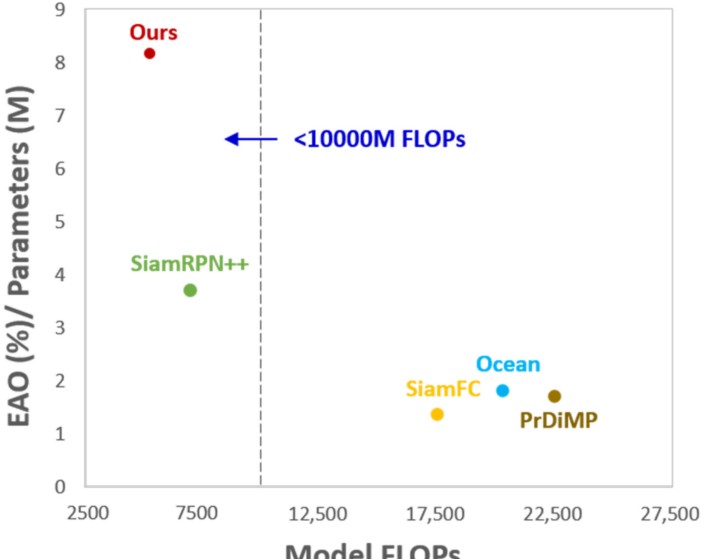

**Figure 4.** Comparisons of different trackers in terms of parameters, FLOPs, and EAO values on VOT2018. The horizontal coordinate represents the floating-point operations (FLOPs) of the model, and the vertical coordinates denote the ratio of the EAO value to the number of parameters measured. Our tracker achieves a better result than other current methods.

### 4.6. Results on TrackingNet

The TrackingNet [26] dataset contains more than 30,000 videos in 27 categories, with a larger number of videos and annotations than any previous dataset. Simultaneously, the dataset makes a distinction between the training set and the testing sets, with no overlap

between the two sets. The large-scale video provided by this dataset can effectively alleviate the current problem of insufficient training data in the field of object tracking.

As shown in Table 4, TransT [18] achieves the highest AUC score, PrDiMP [16] gains an AUC score of 0.750, and our model follows closely. The proposed method a the precision of 0.696 and reaches results comparable to PrDiMP [16] for normalized precision.

**Table 4.** Comparisons with several trackers in terms of AUC score, precision, and normalized precision on TrackingNet. Among them, $P_{norm}$ normalizes the precision and avoids the influence of image scale and the boundary size of the target on the measurement.

| Trackers | AUC | Precision | $P_{norm}$ |
|---|---|---|---|
| ATOM [14] | 0.703 | 0.648 | 0.771 |
| DaSiamRPN [10] | 0.638 | 0.591 | 0.733 |
| DiMP [15] | 0.723 | 0.633 | 0.785 |
| ECO [6] | 0.554 | 0.492 | 0.618 |
| MDNet [20] | 0.606 | 0.565 | 0.705 |
| PrDiMP [16] | 0.750 | 0.691 | 0.803 |
| RT-MDNet [22] | 0.584 | 0.533 | 0.694 |
| SiamFC [7] | 0.571 | 0.533 | 0.663 |
| SiamRPN++ [11] | 0.733 | 0.694 | 0.800 |
| TransT [18] | **0.814** | **0.803** | **0.867** |
| Ours-conf | 0.705 | 0.672 | 0.788 |
| Ours | 0.738 | 0.696 | 0.802 |

*4.7. Comparisons on UAV123*

The UAV123 [27] dataset consists of 123 videos captured by drones, with 9 categories and 12 attributes. The dataset is aimed at specific drone scenes. The videos from this dataset are captured from an overhead perspective with a clean background and rich perspective changes. The results are shown in Table 5. Our method obtains a result comparable to the results of state-of-the-art trackers. There is a substantial improvement in our tracker compared with MDNet [20] and RT-MDNet [22].

**Table 5.** Comparisons with current trackers on the UAV123 and NfS datasets in terms of AUC score.

| Trackers | ATOM [14] | C-COT [5] | DiMP [15] | ECO [6] | MDNet [20] | PrDiMP [16] | RT-MDNet [22] | SiamFC [7] | SiamRPN++ [11] | TransT [18] | Ours |
|---|---|---|---|---|---|---|---|---|---|---|---|
| UAV123 | 0.642 | 0.513 | 0.653 | 0.522 | 0.528 | 0.680 | 0.528 | 0.498 | 0.613 | **0.691** | 0.648 |
| NfS | 0.584 | 0.488 | 0.620 | 0.466 | 0.429 | 0.635 | - | - | 0.502 | **0.657** | 0.632 |

*4.8. Comparisons on NfS*

The NfS [28] dataset includes 100 videos with 17 categories and 9 attributes. Videos from the NfS dataset reach 240 FPS per second. Higher frame rates tend to improve the performance of trackers. The tracking results are also shown in Table 5. Our method can achieve a result similar to PrDiMP [16].

*4.9. Visualization Result*

Additionally, we visualize the tracking results compared with C-COT [5], DaSiamRPN [10], DiMP [15], MDNet [20], SiamFC [7], and SiamRPN++ [11] on several video sequences from OTB100 [24]. Some challenging examples are shown in Figure 5. The proposed method can still effectively track objects under the influence of occlusion, motion blur, illumination, scale change, etc.

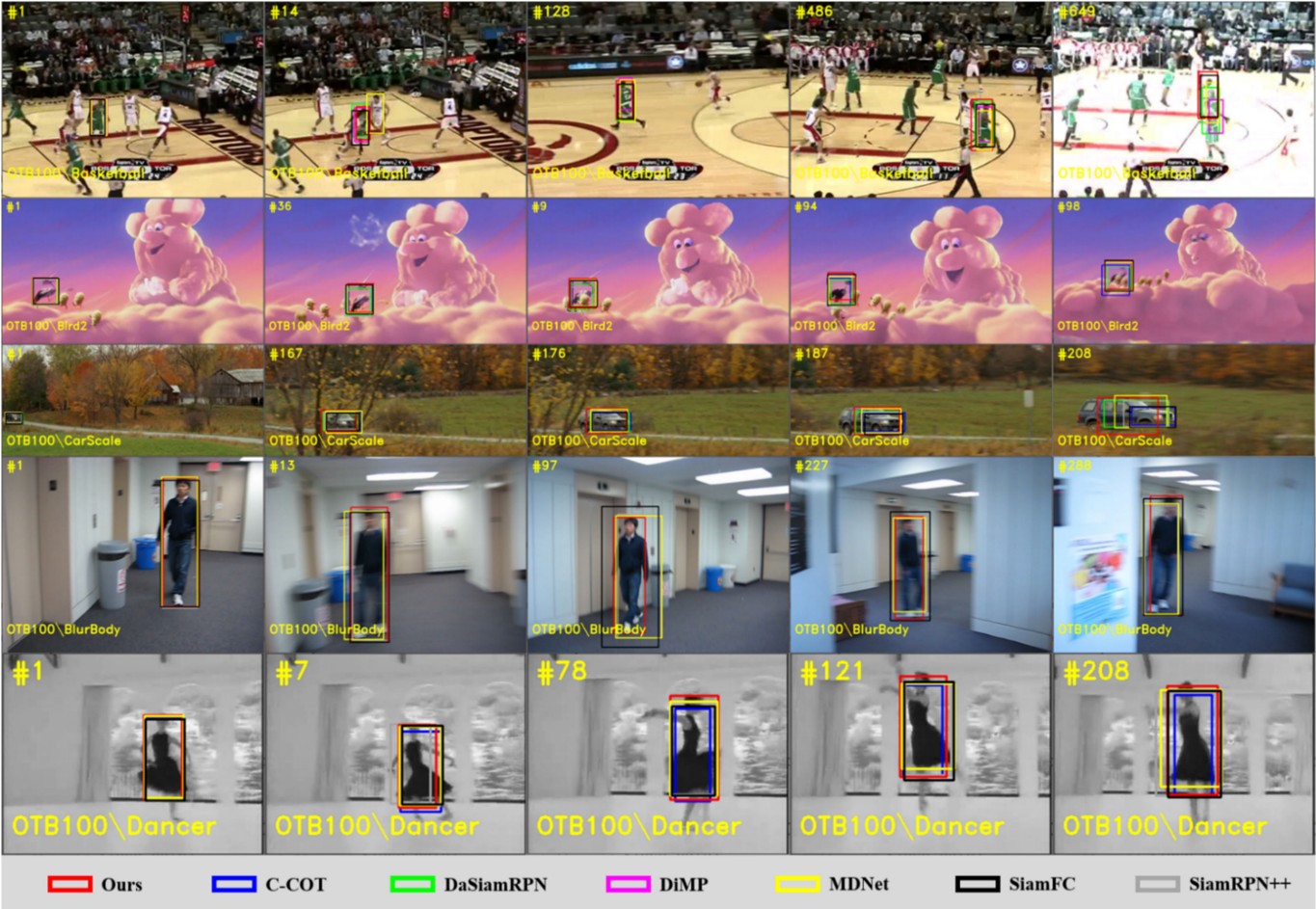

**Figure 5.** Visualization comparisons of the proposed method with state-of-the-art trackers on five video sequences: Basketball, Bird2, CarScale, BlurBody, and Dancer. Our tracker can robustly track the target in each video sequence.

## 5. Discussion

In this paper, we show that Faster MDNet can maintain a better balance among tracking accuracy, tracking speed, and model complexity. The results may be attributed to the following facts. The feature extraction part of Faster MDNet is smaller than other current trackers, containing only three convolutional layers. The channel attention module uses a negligible number of parameters to improve tracking accuracy and maintain tracking speed by capturing inter-channel information efficiently. The number of parameters in the adaptive spatial pyramid pooling layer is much smaller than that of the fully connected layers of MDNet. Therefore, using the adaptive spatial pyramid pooling layer can dramatically accelerate the tracking speed.

However, some limitations of this study still exist. Due to a huge amount of candidate samples requiring feature extraction, Faster MDNet could only run at a speed of 36 FPS, barely reaching real-time tracking. Moreover, to achieve a better balance, our tracker loses tracking accuracy to some extent. Finally, the results cannot prove that Faster MDNet can solve the occlusion problem as effectively as our network.

In summary, we will focus on accelerating our tracker and improving the tracking accuracy in complicated scenes with occlusions in our future work. We will utilize the idea of dynamic adaptive distribution from domain adaptation techniques. Specifically, we will dynamically reduce the distance between the distribution of the features when the target is occluded and when it is not occluded. Subsequently, the model will be more robust to track the object.



## 6. Conclusions

In this paper, a novel model based on MDNet is proposed to resolve the problem that a robust tracker would not only have a high tracking accuracy but also a fast tracking speed with less computational resource consumption. In our work, the channel attention module is implemented behind convolutional layers to adaptively recalibrate the weights of each channel. We design domain adaptation components to extract common features shared by the image classification and object tracking domains. The adaptive spatial pyramid pooling layer is introduced to effectively reduce the number of parameters and to avoid overfitting. The experimental results demonstrate the effectiveness of the proposed method.

**Author Contributions:** Conceptualization, Q.Y.; formal analysis, Q.Y.; investigation, Q.Y.; methodology, Q.Y. and Y.Z.; project administration, Y.Z.; software, Q.Y.; supervision, Y.Z.; validation, Q.Y., K.F., Y.W. and Y.Z.; visualization, Q.Y.; writing—original draft preparation, Q.Y.; writing—review and editing, Q.Y. and Y.Z. All authors have read and agreed to the published version of the manuscript.

**Funding:** This research was funded in part by the National Natural Science Foundation of China under Grants U20B2065, 61972206, and 62011540407; in part by the Natural Science Foundation of Jiangsu Province under Grant BK20211539; in part by the 15th Six Talent Peaks Project in Jiangsu Province under Grant RJFW-015; in part by the Qing Lan Project; and in part by the PAPD fund.

**Data Availability Statement:** Not applicable.

**Conflicts of Interest:** The funders had no role in the design of the study; in the collection, analyses, or interpretation of data; in the writing of the manuscript, or in the decision to publish the results.

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
