# Peer review of "Faster MDNet for Visual Object Tracking"

_applsci, doi:10.3390/app12052336_

Round 1
Reviewer 1 Report
This paper presents a light-weight network for visual object tracking. The proposed method achieves improved tracking speed by proposing domain adaptation components and a spatial pyramid pooling layer. To further improve tracking accuracy, an attention module is designed. Experimental results show the effectiveness of this model. However, the problems of the paper are still apparent:
1. As an extension of a conference paper, the contributions of this paper is slightly insufficient. Domain adaptation components and the spatial pyramid pooling layer have been proposed in the conference version.
2. This paper has limited novelty by integrating attention module, spatial pyramid pooling layer into a holistic model.
3. Transformer-based tracking methods mentioned in the introduction of the paper, which currently achieve state-of-the-art results, do not appear in the experiment part.
4. The method in the pre-expansion conference paper lacks a performance comparison with the method in this paper in the experiment part.
5. The proposed method has achieved certain improvements, however, it is still only close to the existing methods. The authors can investigate this problem further.
Reviewer 2 Report
The Authors propose a Faster MDNet version, specifically modified for improving tracking accuracy and speed while reducing computational complexity. Results presented demonstrate how the changes implemented have made it possible the achievement of objectives set.
The overall quality of the paper is sound, and well described. Consequently, no revisions are required, but only an optional comment is provided hereafter.
The conclusion section is quite limited if compared to the kind of work described and results achieved. I would therefore suggest to deepen this section a little, describing in more detail the conclusions by retracing the entire work and future directions of the developed solution.
Reviewer 3 Report
Authors in this manuscript identify the current state-of-the-art object tracking methods maybe improved. In general, this work is well motivated and the authors have done comprehensive review and analyses for existing works where they identified problems such as in computing costs, speed, and accuracy.
Yet it can be improved:
1) In terms of evaluation experiments, it would be better to have similar or comparable metrics for all datasets to be better demonstrate the conclusion, especially considering Table 5. Currently mixed results can be observed despite overall the proposed method is overall merit.
2) It would be good if you can explain Figure 2 why your ECA can be comparably unstable as Kernel size increasing.
3) More insights on the proposed method can be expected, it shall be more than a simple introduction of how the workflow is.
4)The introduction can be lengthy and may be reduced, especially the paragraphs states related works. Briefs should be okay as details are in the related works.
5) the paragraph before list of contributions can be redundant while what aspects your proposed methods will improved and briefly how will that work are expected.
6) It would be expected to give better statements on the datasets and metrics the tests are performed on.
7) Some minor typos: The commas right after equation 7 and 8 can be confusing; Line 300 the GTX shall be RTX;
Reviewer 4 Report
This paper proposes the approach to faster MDNET for OT. I have some comments as follows.
- Fig.1 : Please change the fonts for readability. Also, in the figure, please insert some more parameters that make the figure more specific.
- Can you please add more comparison results in Table 1 and 2?
- How did you calculate accuracy in Table 3? Please write definition.
- Based on the results in Tables, your results are better than any other results. I think that there are some other work that is superior to yours in some different perspectives. Please explain that once you find ones that show better results than yours.
- Please add discussion about your results.
Round 2
Reviewer 1 Report
This paper can be accepted.